# Ecological Mapping in Assessing the Impact of Environmental Factors on the Aquatic Ecosystem of the Arys River Basin, South Kazakhstan

**Elena G. Krupa** [1] , **Sophia S. Barinova** [2,*] **and Sophia M. Romanova** [3]

[1]  Institute of Zoology, Ministry of Education and Science, Science Committee, Almaty 050060, Kazakhstan; elena_krupa@mail.ru
[2]  Institute of Evolution, University of Haifa, Mount Carmel 3498838, Israel
[3]  Ministry of Education and Science, Al-Farabi Kazakh, National University, Almaty 050040, Kazakhstan; sofya.romanova@kaznu.kz
*  Correspondence: sophia@evo.haifa.ac.il; Tel.: +972-4824-97-99

**Abstract:** Assessment of the water quality of the Arys River basin based on the spatial distribution of richness of phytoperiphyton communities and abiotic variables was given for the first time. Altogether, 82 species were revealed in phytoperiphyton, including Bacillariophyta of 51, Cyanobacteria of 20, Chlorophyta of 7, and Charophyta of 4. Cluster analysis revealed the uniqueness of the composition of periphyton communities related to the abiotic conditions. The environmental preferences of the algae indicated fresh organic pollution in the lower reaches of the Arys River and weak or moderate levels of organic pollution in the rest of the basin. The ecological mapping of chemical data generally confirmed this conclusion. According to the maps, the highest water quality was revealed in the upper stream of the basin. The middle part of the river basin had the lowest water quality in terms of transparency, nitrite-nitrogen, and nitrate-nitrogen. The downstream of the Arys was characterized by a secondary deterioration in water quality according to the Aquatic Ecosystem State Index (WESI) index. We revealed the complicated interaction between natural and anthropogenic factors that caused changes in water quality in the Arys River basin.

**Keywords:** phytoperiphyton; river; bioindication; ecological mapping; water quality

## 1. Introduction

High human population density, along with a large number of industrial and agricultural enterprises, determines the pronounced anthropogenic pressure on the water bodies of Southern Kazakhstan [1–3]. An additional factor affecting the quality of water in water sources is mining in the region [4].

The Syr Darya, the largest river in southern Kazakhstan, begins in the Tien Shan mountains. The upper sections of the river, along with the main tributaries, are located far from sources of anthropogenic impact. Like other mountain water bodies [5,6], the mountainous part of the Syr Darya basin is characterized by high-quality water. The plains of the river basin are prone to contamination caused by the use of water in industry, utilities, agriculture, and livestock [7,8].

Water-quality problems in southern Kazakhstan wholly relate to the Arys River [8]. This largest tributary of the Syr Darya originates in the mountains of Talas Alatau. The analysis of the published data showed that assessment of the ecological state of the Arys River has not yet been carried out; however, in the references, hydrochemical variables [9] and the composition of the periphyton algae are given [10]. A few more publications are devoted to the periphyton [11] or phytoplankton communities [12–14] of some reservoirs in Southern Kazakhstan.

According to the European Water Framework Directive [15], water quality assessment should be based on biological and physicochemical parameters. The harsh conditions of mountain rivers (rapid flow, rocky bottom, low water temperatures) are unfavorable for the development of phytoplankton [16] and zooplankton [17]. As a result, planktonic communities are used only for assessment of water quality of small mountain rivers with slow flow [18]. Periphyton communities do not drift because of the current, and therefore are good indicators of changes in the water quality of mountain rivers [16,19,20] like Arys and its tributaries.

Under current trends [21,22], water-quality assessment should take into account the river basin principle, which considers the aquatic ecosystem as an accumulative part of the basin. The intensity of land use affects the number of nutrients entering the water bodies and the quality of the water [23]. Ecological mapping [24] fully complies with the basin principle, since it allows us to estimate the contribution of terrestrial territories to the total pollution of river ecosystems.

Following international experience [15,21,25,26], this work is the first to provide a comprehensive assessment of the ecological state of the Arys River basin based on the ecological mapping of biotic and abiotic data. For the water quality assessment, we used the composition of phytoperiphyton communities, the saprobity index SI, the Aquatic Ecosystem State Index WESI index, and also physical and chemical data like water transparency, temperature, total dissolved solids, the content of nitrite-nitrogen, nitrate-nitrogen, and ammonium-nitrogen.

## 2. Description of the Study Area

The climate of the region is continental. Winter is moderately warm, with thaws of up to +10 °C and cold snaps to −15 °C. Summers are hot, long, with a maximum air temperature of up to +49 °C. The average annual precipitation is 100–200 mm, in mountainous areas up to 1600 mm [27].

The Arys River is the largest right tributary of the Syrdarya River in South Kazakhstan (Figure 1). It starts from springs on the northwestern slopes of Talas Alatau at an altitude of about 1513 m. The length of the river is 378 km. The average water consumption is 46.6 m$^3$/s. Water is used for irrigation, so only a small part of it reaches the Syr Darya River. Water intake from Arys is carried out by 37 canals, the largest of which is the Arys–Turkestan canal. The width of the river reaches 40–50 m in the lower reaches, with a floodplain width of 1.5–2 km. The river banks are represented by yellow and red clays in the middle and lower reaches. The coasts mostly are overgrown with bushes and trees.

The left tributaries of the river are the Aksu, Sairamsu, Jetimsay, Jabaglysu. They originate on the slopes of Talas Alatau at the altitude of 1513–2200 m. The length of the Aksu River is 133 km, and the basin area is 766 km$^2$. It flows through a narrow valley in its upper reaches. The width of the river reaches 40–50 m in the lower part. Sairamsu, Jetimsay, Zhabaglysu are second-order tributaries. Their length does not exceed 36–76 km. Another left tributary of Arys is Badam, which originates in the mountains of Karzhantau. The length of the Badam River is 141 km. Its width is 10–15 m, its depth is 0.5–2.0 m. The area of its basin is 4.3 km$^2$. A large number of settlements are located in the middle reaches of the river, where intensive economic activity is carried out. The right tributaries start from the spur in the Boralday ridge (Karatau Mountains). The banks of the tributaries of the Arys River in the upper reaches are mostly stony, and sometimes they are composed of loess-like loam. They mostly are overgrown with bushes and trees. The area of the Arys River basin, together with its tributaries, is 14,900 km$^2$.

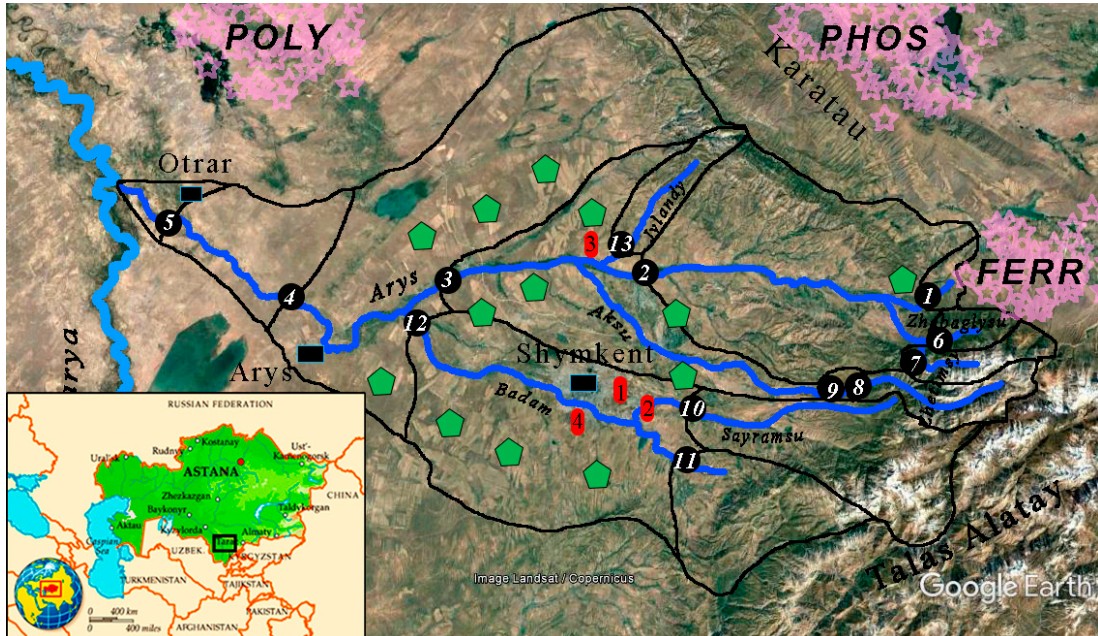

**Figure 1.** The Arys River basin in Southern Kazakhstan with sampling stations and pollution points, June 2016. Black squares are cities; Black circles with white numbers are sampling stations; Green polygons are agricultural fields. Red polygons are sources of industrial pollution: 1—Industrial enterprises of Shymkent, 2, 3—Cement plants, 4—Industrial waste dump. Deposits: POLY—polymetallic ores, PHOS—phosphorites, FERR—iron ores.

## 3. Sources of Pollution

The territory of the Arys River basin is one of the most densely populated in Kazakhstan [26]. It is used to grow cereals, cotton, rice, vegetables, grapes, and as pastures for sheep and camels (Figure 1). The region is rich in deposits of minerals [1]. Phosphorites, rare earth, and fluorine are mined in the Karatau Mountains (Figure 1, PHOS). There are deposits of polymetallic (Figure 1, POLY) and iron ores (Figure 1, FERR) in the river basin. Mineral processing industrial enterprises are located in the cities of Shu, Arys, Karatau, as well as in small towns (Figure 1, Red polygons).

Eleven reservoirs and ponds, as well as the Arys–Turkestan canal, were created to provide water for agriculture and industry in the Arys River basin. One of the sources of pollution of terrestrial areas and water resources of the region are industrial (Figure 1) and the rubbish dumps [3].

## 4. Material and Methods

Phytoperiphyton samples were collected from 13 stations in June 2016. A total of 28 samples from different substrates (stones, water plants) were taken. Phytoperiphyton samples were collected taking into account visible differences in submerged substrates. Two samples of phytoperiphyton most often were collected from each station. The algal samples were obtained by scratching of periphyton, put to the 15 mL plastic tubes, and fixed in 4% of neutral formaldehyde solution [28]. Water samples were taken at the same stations to determine the Total Dissolved Solids (TDS), nitrites, nitrates, ammonia and easily oxidized organic substances in water (permanganate index, PI) [29]. Water samples for nutrient analysis were taken in 0.5 L glass containers and fixed by adding 1 mL of chloroform. Water samples for determination of Permanganate Index (PI) were taken in 0.250 mL glass containers and fixed by adding chemically pure sulfuric acid at a 1:3 dilution. All collected samples were delivered to the laboratory in the refrigerator. The generally accepted methods of chemical analysis of water were used [30,31]. The temperature and pH values of the surface layers of water were measured in parallel with sampling by HANNA HI 9813-0. The transparency of the water was measured using a Secchi disk. Coordinate referencing of the stations was done by Garmin eTrex GPS-navigator.

Fixed phytoperiphyton samples were studied in three repetitions in wet and permanent slides [32] under light microscopes Nikon ECLIPSE E200, with a magnification of ×100–×2000. Species identification of algae was performed by using handbooks for relevant taxonomic divisions [33–37]. We calculated the Saprobic Index (SI) introduced by V. Sládeček [38] in regard to Pantle and Buck's [39] method to estimate the level of organic pollution. Index values SI ranges from 1 to 4 for aquatic environments. All data were ranked according to the classification system from an environmental perspective (Table 1), adopted in the CIS (Former Soviet Union) countries [40,41], to assess Arys basin water quality further.

**Table 1.** Classification of the quality of natural waters by abiotic variables and the saprobity index SI according to [40,41].

| Water Quality Class | Rank | Color Code | N-NH$_4^+$ mg/L | N-NO$_2^-$ mg/L | N-NO$_3^-$ mg/L | PI, mg O/L | Secchi Transparency, m | Sládeček's Saprobity SI |
|---|---|---|---|---|---|---|---|---|
| I—very pure | 1 | blue | <0.05 | 0 | <0.05 | <2.0 | >3.00 | <0.5 |
| II—pure | 2 | green | 0.05–0.10 | 0.001–0.002 | 0.05–0.20 | 2.0–4.0 | 0.75–3.00 | 0.5–1.0 |
| II—pure | 3 | green | 0.11–0.20 | 0.003–0.005 | 0.21–0.50 | 4.1–6.0 | 0.55–0.70 | 1.0–1.5 |
| III—moderate | 4 | yellow | 0.21–0.30 | 0.006–0.010 | 0.51–1.00 | 6.1–8.0 | 0.45–0.50 | 1.5–2.0 |
| III—moderate | 5 | yellow | 0.31–0.50 | 0.011–0.020 | 1.01–1.50 | 8.1–10.0 | 0.35–0.40 | 2.0–2.5 |
| IV—polluted | 6 | orange | 0.51–1.00 | 0.021–0.050 | 1.51–2.00 | 10.1–15.0 | 0.25–0.30 | 2.5–3.0 |
| IV—polluted | 7 | orange | 1.01–2.50 | 0.051–0.100 | 2.01–2.50 | 15.1–20.0 | 0.15–0.20 | 3.0–3.5 |
| V—very polluted | 8 | red | 2.51–5.00 | 0.101–0.300 | 2.51–4.00 | 20.1–25.0 | 0.05–0.10 | 3.5–4.0 |
| V—very polluted | 9 | red | >5.00 | >0.300 | >4.00 | >25.0 | <0.005 | >4.0 |

We calculated the WESI index [42,43] using the formula: WESI = Rank Index SI/Rank N-NO$_3$ to assess the toxic pollution influence to the aquatic ecosystems of the Arys basin. The index values vary from 0 to 5. If the index values are less than unity, then the ecosystem is exposed to toxic pollution, which inhibits photosynthesis. The calculation of the similarity of the species composition of algal communities at river stations based on the Sørensen indices and the construction of the dendrogram were carried out using the Biodiversity Pro program [44].

A new method of spatial mapping construction [24] was used for the analysis of the relationship between biological and environmental variables. The mapping was based on the postulate, which treats any water body as an accumulative part of the basin. Water flows from top to bottom along with the relief of the basin, so certain and measured parameters of water quality at each station reflect the sum of the processes that take place on the watercourse and basin from this point to the next station.

## 5. Results

### 5.1. Environmental Variables

The Arys River basin is located at different altitudes, which change from 251 to 1513 m (Table 2). The water temperature ranged from 10.2 to 27.0 °C during the sites. The maximum water temperature was revealed in the downstream of the Arys River and its tributaries. At pH values of 7.5–8.9, the rivers in the upper stream had more alkaline water relative to the plain areas. The water depth varied from 0.1 to 2.0 m. The water transparency was very low. The color of the water was changing from brown (st. 1) to clayey (st. 4, 5) in the different parts of the Arys River. The color of the water in the tributaries varied from colorless (st. 7 and spring) to various shades of gray and green (st. 5–13).

The TDS varied from 160.1 to 526.0 mg dm$^{-3}$ (Table 3). The minimum values of TDS were found in the mountainous areas of the Arys basin. The highest values of TDS were revealed in the lower reaches of the Arys River, and the Jilandy River. The content of nitrite-nitrogen varies from 0.002 to 0.033 mg/L, with the maximum values in the Arys (st. 3 and 4), Sayramsu (st. 10), and Badam Rivers

(st. 11). The maximum content of nitrate-nitrogen was revealed in Arys as well as in Badam and Jilandy rivers. The concentrations of ammonium-nitrogen were higher in the tributaries and the spring than in the Arys River. The highest values of PI were noticed in the upper part of Arys and its right tributary Jilandy.

**Table 2.** Sampling stations on the Arys River basin with coordinates and major environmental variables, June 2016.

| River Name | No of Station | North | East | Altitude, m a.s.l. | Depth, m | Secchi Transparency, m | Tempe-rature, °C | Color of Water |
|---|---|---|---|---|---|---|---|---|
| Arys | 1 | 42°30′37.0 | 70°37′14.6 | 1135 | 0.10–0.15 | 0.05 | 22.2 | brown |
| Arys | 2 | 42°34′50.91 | 69°58′20.57 | 494 | 0.7–0.8 | 0.2 | 23.1 | greenish |
| Arys | 3 | 42°35′13.8 | 69°18′31.6 | 289 | 1.5–2.0 | 0.1 | 23.0 | light brown |
| Arys | 4 | 42°27′52.2 | 69°57′02.5 | 231 | 1.0–1.5 | 0.1 | 24.9 | clayey |
| Arys | 5 | 42°41′16.26 | 68°27′13.18 | 205 | 1.5–2.0 | 0.2 | 27.2 | clayey |
| Jabaglysu | 6 | 42°25′11.3 | 70°33′00.7 | 1330 | 1.3–1.4 | 0.1 | 13.9 | light gray |
| Jetimsay | 7 | 42°24′18.7 | 70°32′50.1 | 1513 | 0.3–0.4 | 0.3–0.4 | 10.2 | colorless |
| Aksu | 8 | 42°20′06.3 | 70°27′10.1 | 1469 | 1.5–1.8 | 0.2 | 12.2 | whitish |
| *spring | 9 | 42°20′06.3 | 70°27′10.1 | 1469 | 0.10–0.15 | to the bottom | 10.8 | colorless |
| Sayramsu | 10 | 42°15′50.14 | 69°57′22.96 | 873 | 1.5 | 0.2 | 14.5 | light brown |
| Badam | 11 | 42°06′02.7 | 69°57′48.2 | 960 | 0.7–0.8 | 0.7–0.8 | 15.6 | light gray |
| Badam | 12 | 42°30′08.6 | 69°04′14.2 | 251 | 1.5–2.0 | 0.1 | 23.2 | clayey |
| Jilandy | 13 | 42°35′56.2 | 70°14′25.3 | 723 | 0.2–0.3 | 0.2–0.3 | 17.0 | greenish |

Note. *—The transparency of the water was equal to the depth of the spring, i.e., to the bottom.

**Table 3.** Chemical variables of the Arys River and its tributaries and the saprobity index, June 2016.

| RIVER NAME | No of Station | TDS, mg/L | PI, mg O/L | $N\text{-}NO_2^-$, mg/L | $N\text{-}NO_3^-$, mg/L | $N\text{-}NH_4^+$, mg/L | Index SI |
|---|---|---|---|---|---|---|---|
| Arys | 1 | 215.3 | 4.62 | 0.018 | 0.443 | 0.039 | 1.82 |
| Arys | 2 | 382.3 | 3.00 | 0.008 | 2.060 | 0.008 | 1.86 |
| Arys | 3 | 410.2 | 2.00 | 0.030 | 2.115 | 0.023 | 1.84 |
| Arys | 4 | 440.7 | 3.20 | 0.008 | 0.000 | 0.023 | 1.91 |
| Arys | 5 | 526.0 | 2.83 | 0.033 | 1.267 | 0.016 | 1.74 |
| Jabaglysu | 6 | 160.1 | 2.60 | 0.002 | 0.319 | 0.031 | 2.03 |
| Jetimsay | 7 | 239.9 | 2.20 | 0.009 | 0.001 | 0.023 | 1.74 |
| Aksu | 8 | 164.6 | 2.08 | 0.006 | 0.233 | 0.039 | 1.62 |
| spring | 9 | 198.2 | 1.72 | 0.002 | 0.002 | 0.070 | 1.56 |
| Sayramsu | 10 | 161.8 | 2.22 | 0.032 | 0.373 | 0.054 | 2.11 |
| Badam | 11 | 187.4 | 2.24 | 0.030 | 0.194 | 0.039 | 1.72 |
| Badam | 12 | 415.6 | 3.02 | 0.010 | 1.936 | 0.047 | 1.62 |
| Jilandy | 13 | 521.5 | 4.44 | 0.005 | 2.946 | 0.047 | 1.95 |

*5.2. Characteristics of Phytoperiphyton*

Altogether 82 taxa of algae and cyanobacteria were revealed in phytoperiphyton. The Bacillariophyta was the richest with 51 species, including *Nitzschia* with 9 taxa and *Cymbella* with 6 taxa as the richest genera. Cyanobacteria were represented by 20 taxa, Chlorophyta by 7, and Charophyta by 4 taxa. Green and charophyte algae enriched the periphyton communities at the

altitude of about 200 m a.s.l. (Figure 2). At the altitudes up to 1000 m a.s.l., 36 algae species were revealed. The most common species were from diatoms *Cymbella*, *Diatoma*, and *Encyonema* as well as planktonic or planktonic-benthic cyanobacteria from *Phormidium*, *Planktolyngbya*, and *Pseudanabaena*. At the altitudes of about 1500 m, there were mostly diatoms of the genera *Diatoma*, *Encyonema*, *Cymbella*, *Nitzschia*, and *Surirella*.

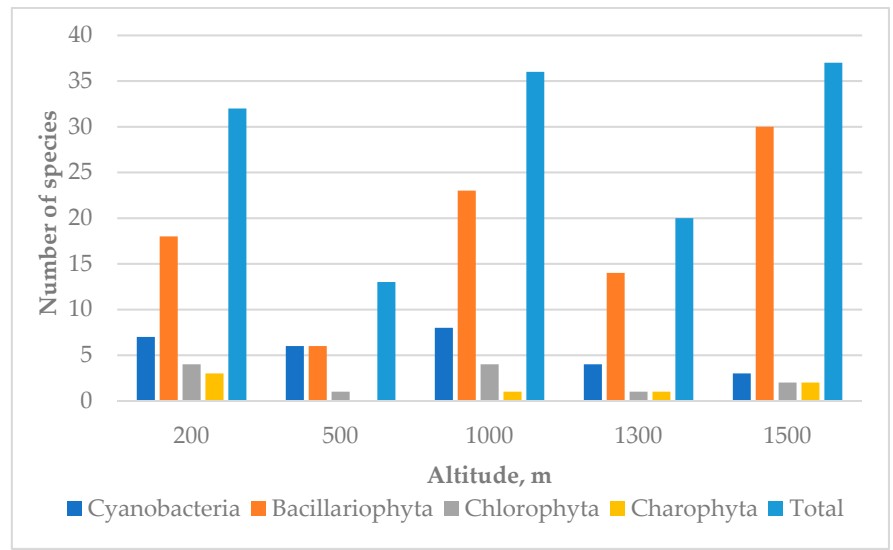

**Figure 2.** Distribution of species richness in the Arys River communities over altitude of habitats.

Cluster analysis revealed the uniqueness of the species composition of the periphyton communities of the different parts of the Arys River basin (Figure 3). At a 50% similarity level, the communities of each station were singled out in a separate cluster, except for two stations in the lower reaches of the Arys River. With a similarity level of about 40%, periphyton communities formed 6 clusters.

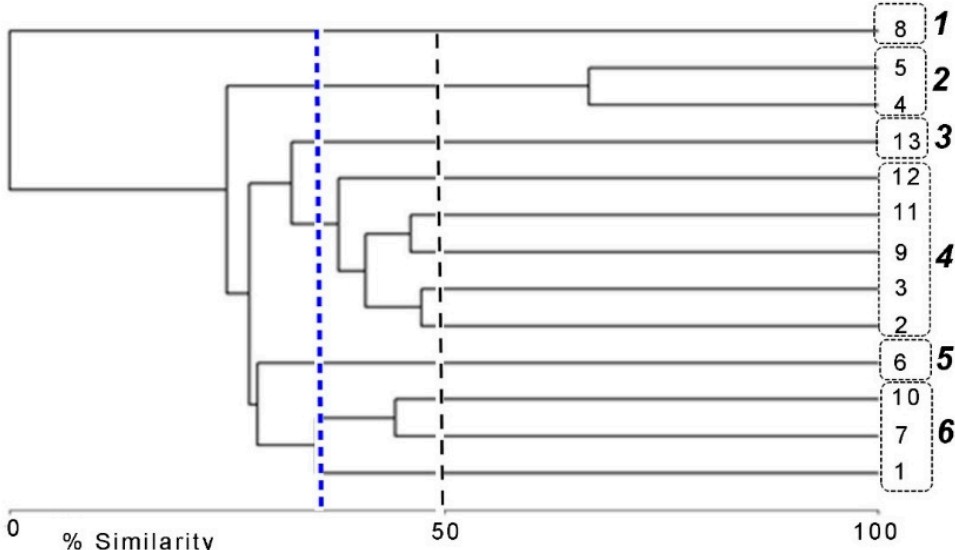

**Figure 3.** Dendrogram of the similarity of the species composition of periphyton in the examined parts of the Arys River basin, June 2016. The numbers and descriptions of the clusters are given in Table 4.

**Table 4.** The toxic pollution index Aquatic Ecosystem State Index WESI and ranks (numbers) and classes (color codes) of water quality in the classification of hydrochemical variables of the Arys river basin from an ecological point of view in the CIS (Former Soviet Union) countries, June 2016.

| No of Station | Transparency | PI | $N\text{-}NO_2^-$ | $N\text{-}NO_3^-$ | $N\text{-}NH_4^+$ | Index SI | WESI |
|---|---|---|---|---|---|---|---|
| 1 | 5 | 3 | 5 | 3 | 1 | 4 | 1.33 |
| 2 | 7 | 2 | 4 | 7 | 1 | 4 | **0.57** |
| 3 | 8 | 2 | 6 | 7 | 1 | 4 | **0.57** |
| 4 | 8 | 2 | 4 | 1 | 1 | 4 | 4.00 |
| 5 | 7 | 2 | 6 | 5 | 1 | 4 | **0.80** |
| 6 | 8 | 2 | 2 | 3 | 1 | 5 | 1.67 |
| 7 | 5 | 2 | 4 | 1 | 1 | 4 | 4.00 |
| 8 | 7 | 2 | 4 | 3 | 1 | 4 | 1.33 |
| 9 | 7 | 1 | 2 | 1 | 2 | 4 | 4.00 |
| 10 | 7 | 2 | 6 | 3 | 2 | 5 | 1.67 |
| 11 | 2 | 2 | 6 | 2 | 1 | 4 | 2.00 |
| 12 | 8 | 2 | 4 | 6 | 1 | 4 | **0.67** |
| 13 | 6 | 3 | 3 | 8 | 1 | 4 | **0.50** |

Note. Class of Water Quality in European Union (EU) color code: Class 1, very pure—blue, Class 2, pure—green, Class 3, moderate—yellow, Class 4, polluted—orange, Class 5, very polluted—red. Index WESI color code: light lilac—the value more than 1, dark lilac—the value less than 1.

The uniqueness of the species composition of the periphyton communities is due to the variety of conditions in the Arys River basin. The analysis showed that fouling complexes consisted mainly of benthic and plankton-benthic species (Figure 4). Under conditions of a slower course of down parts of the basin (st. 5,8,9,10), the proportion of planktonic species in communities increased. Indicators of temperature conditions are represented mainly by moderately thermophilic species. In the middle course of the Arys River (st. 2, 3), as well as in the rivers Aksu, Badam, and Zhilanda with a relatively slow stream (st. 8, 11, 13), the proportion of thermophilic species increased. Regarding oxygen conditions, species that were indicators of weakly and moderately saturated oxygenated waters prevailed. Aerophiles that preferred conditions with a high oxygen content were noted only in the phytoperiphyton community of the Aksu River (st. 8). Indifferent species inhabiting a wide range of salinity prevailed in all studied communities. In the Arys River (st. 1–5), the phytoperiphyton communities were enriched by mesohalobes, which prefer waters with a significant saturation with chlorides.

Indicators of neutral or slightly alkaline waters prevailed in the phytoperiphyton communities of the studied basin (Figure 5). The ecological preferences of algae species indicated alkaline conditions in the Arys River. Alkaliphiles and alkalibionts comprised from 25 to 55% of the species composition of communities in the remaining parts of the basin. Watanabe organic pollution indicators included only diatom species. Indicators of pure water accounted for from 20 to 65% of phytoperiphyton communities. According to the change in the proportion of species—indicators of polluted waters, the organic pollution of the Arys River water increased downstream. Some of the algae species belonged to saprophiles (st. 7, 11, 12) like *Nitzschia gracilis*, *N. palea*. The proportion of eurysaprobes varied from 25 to 70%. According to the values of the saprobity index SI, indicators of moderately polluted waters prevailed in the composition of all periphyton communities. The proportion of oligotrophic and mesotrophic species increased from the upper to the lower sections of the Arys River. At the same time, in the lower reaches of the river (st. 4, 5), there was a noticeable proportion of hypereutrophic species. The waters of the tributaries Jabaglysu, Jetimsay, Sayramsu, Jilandy, and the average course of the Badam River (st. 6, 7, 10, 11, 13) can be estimated as mesotrophic. In the lower reaches of the Badam River (st. 12), the presence of hypereutrophic species is noticeable. Oligotrophic species prevailed only in the community of the Aksu River (st. 8).

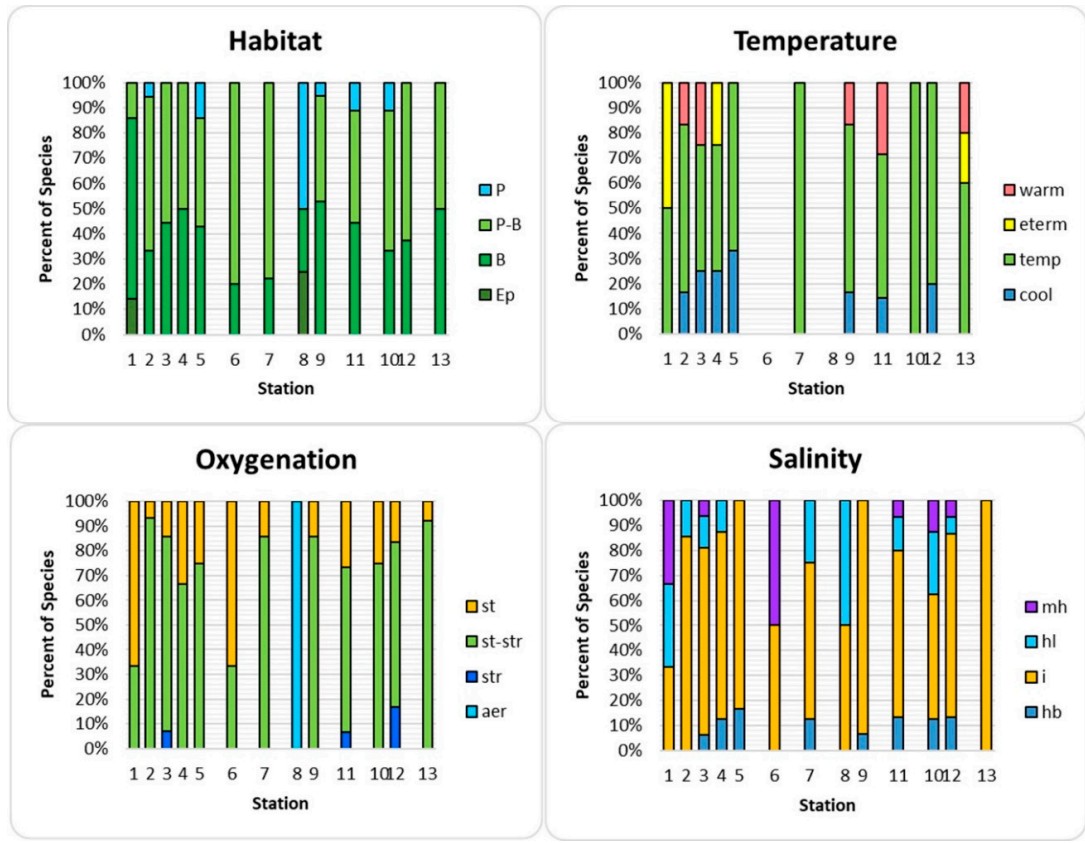

**Figure 4.** Distribution phytoperiphytonic species which are indicators of habitat, temperature, oxygenation, and salinity in the Arys River basin.

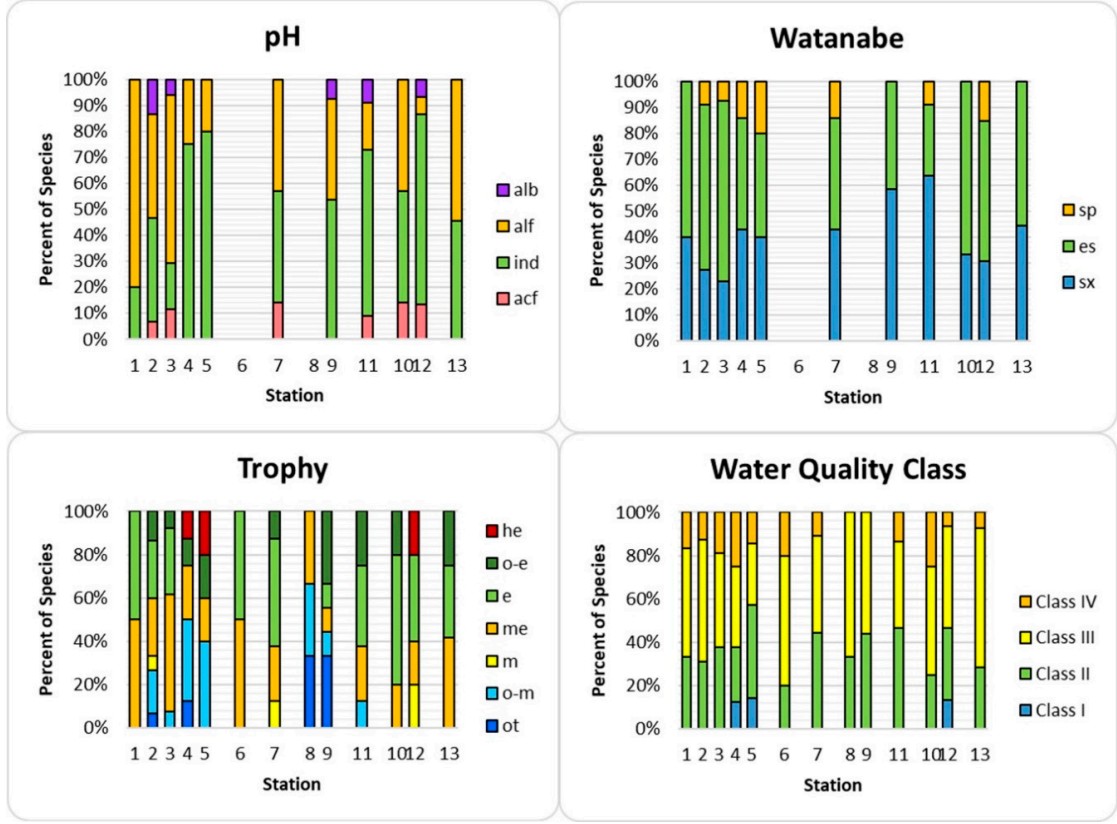

**Figure 5.** Distribution of phytoperiphytonic species which are indicators of pH, of organic pollution according Watanabe, trophic state and water quality class in the Arys River basin.

### 5.3. Water-Quality Assessment of the Arys River Basin by Abiotic Variables

The primary data (Tables 2 and 3) were ranked according to Table 1. According to the results (Table 4), the transparency of the water in the basin varied from moderately polluted to very dirty waters. PI values characterized the water of the Arys River and its tributaries as slightly polluted or less often as clean. The nitrite-nitrogen content varied within 2–4 water-quality classes, nitrate-nitrogen within 1–4 water-quality classes, and ammonium-nitrogen within 1–2 water-quality classes. The saprobity index SI characterized a moderate level of organic pollution of all rivers. According to the WESI index, some parts of the Arys River basin were under the influence of toxic contaminants that suppressed the photosynthesis of periphyton cells.

### 5.4. Water-Quality Assessment of the Arys River Basin by Biological Variables

Figure 3 shows that clusters 1,5,6 united peripheral communities of mountain areas of the basin. These sites were characterized by a low nitrate-nitrogen content, at levels 1 and 2 of the water-quality class, and the absence of toxic pollution determined by the WESI index (Table 4). The habitat of periphyton communities in the lower and middle parts of the basin (clusters 2–4) was more diverse. These sites were characterized by the presence of toxic pollution according to WESI and a wide range of nitrate-nitrogen content, from pure waters of the 1st quality class to very polluted waters of the 5th quality class.

The composition of the periphyton algae species, which determined the similarity within the clusters, is presented in Table 5. The environmental preferences of the algae indicated fresh organic pollution in the lower reaches of the Arys River (cluster 2), a slightly increased TDS and a moderate level of organic pollution of the Jilandy River (cluster 3), weak or moderate levels of organic pollution in the rest of the basin. This conclusion generally coincided with the spatial distribution of indicator algae species (Section 5.2) and the chemical data given in Table 2, Table 3, and Table 5.

**Table 5.** Species of periphyton algae, which determine the similarity within the clusters at the level of 40%, and some abiotic variables in various parts of the Arys River basin, June 2016.

| Cluster Number | Stations | Name | Common Species | Number of Species | TDS, mg/L | Altitude, m a.s.l. |
|---|---|---|---|---|---|---|
| 1 | 8 | Aksu | *Anathece clathrata* (W.West and G.S.West) Komárek, Kastovsky and Jezberová | 3 | 164.6 | 1469 |
| 2 | 4,5 | Arys, downstream | *Stigeoclonium* sp., *Ulnaria ulna* (Nitzsch) P.Compère, *Nitzschia palea* (Kützing) W.Smith | 10 | 440.7–526.0 | 1469 |
| 3 | 13 | Jilandy | *Ulnaria ulna* (Nitzsch) P.Compère, *Stigeoclonium* sp., *Rhoicosphenia abbreviata* (C.Agardh) Lange-Bertalot | 17 | 521.5 | 723 |
| 4 | 2,3,9, 11,12 | Arys, middle reaches, Badam, spring | *Diatoma vulgaris* Bory, *Mougeotia* sp., *Diatoma hyemalis* (Roth) Heiberg, *Ulnaria ulna* (Nitzsch) P.Compère | 52 | 187.4–415.6 | 251–960, 1469 |
| 5 | 6 | Jabaglysu | *Lyngbya aestuarii* Liebman ex Gomont, *Oscillatoria curviceps* C.Agardh ex Gomont, *Klebsormidium subtile* (Kützing) Mikhailyuk, Glaser, Holzinger and Karsten | 5 | 160.1 | 1330 |
| 6 | 1,7, 10 | Arys, upstream, Jetimsay, Sayramsu | *Encyonema elginense* (Krammer) D.G.Mann, *Pseudanabaena limnetica* (Lemmermann) Komárek, *Navicula cryptocephala* Kützing, *Eolimna minima* (Grunow) Lange-Bertalot and W.Schiller | 19 | 161.8–239.9 | 873–1513 |

### 5.5. Ecological Mapping

Ecological mapping of ranked data (Figure 6) made it possible to identify areas of the basin that are responsible for the variability of the analyzed indicators. The ammonium content throughout the

basin was at the pure water level, locally increasing in the upper part (Figure 6e). The permanganate index (PI) characterized mostly a low amount of organic substances in the water, at the level of class 2 (Figure 6b). According to the variability of transparency (Figure 6a), nitrite-nitrogen (Figure 6c), and nitrate-nitrogen (Figure 6d), water quality deteriorates mainly in the middle course of the Arys River. Minimum transparency at the level of the 5th class of the very polluted waters was noted here. The content of nitrites and nitrates corresponded to the level of 3 classes, with moderately polluted water, and 4 classes with polluted water. It should be noted that there was a local improvement in water quality in the lower reaches of the river according to the last three variables.

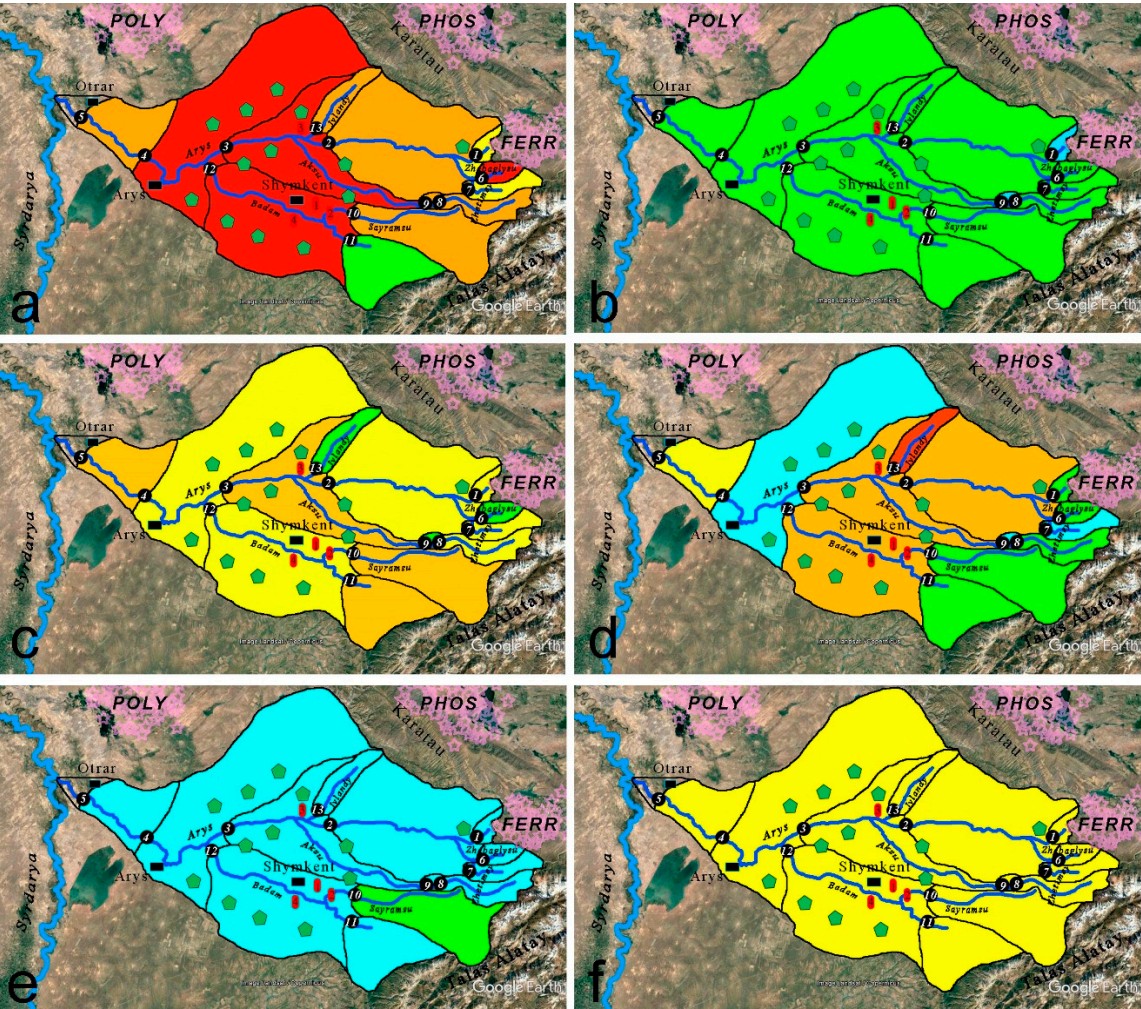

**Figure 6.** Ecological mapping of the Arys River basin. Transparency (**a**), permanganate index (PI) (**b**), nitrite-nitrogen content (**c**), nitrate-nitrogen content (**d**), ammonium-nitrogen content (**e**), the saprobity index SI (**f**). The regions toned in respect of Class of Water Quality in European Union (EU) color code as in Table 4: Class 1, very pure–blue, Class 2, pure–green, Class 3, moderate–yellow, Class 4, polluted–orange, Class 5, very polluted–red.

The saprobity index SI characterized a moderate level of organic pollution of the Arys River basin (Figure 6f). According to the WESI index values, the upper part of the basin did not experience toxic load (Figure 7). The middle reach of the Arys River with its tributaries was exposed to toxic pollution, which adversely affected algal communities. Downstream, toxic pollution decreased and increased again on the site near the confluence of the Arys River into the Syr Darya.

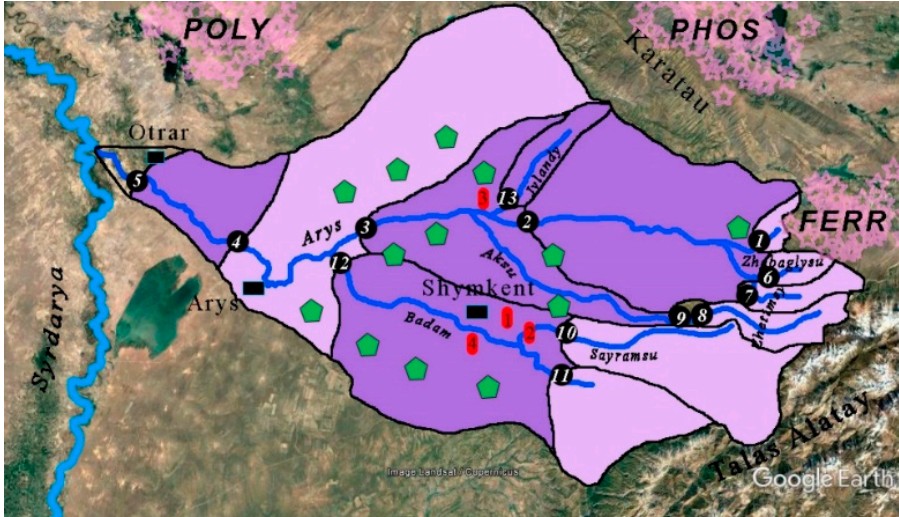

**Figure 7.** Ecological mapping of the Arys River basin according to the WESI index. Index WESI color code as in Table 4: light lilac—the value more than 1, dark lilac—the value less than 1.

## 6. Discussion

According to the results obtained, the periphyton communities of various sections of the Arys River basin had a peculiar composition. Diatoms were the most diverse in the upper parts of the basin, while Cyanobacteria, Chlorophyta, and Charophyta got enriched the periphyton communities of the lower parts of the rivers. Cluster analysis also revealed an indirect relationship between the composition of periphyton and such factors as the altitude-climatic gradient, nutrient content, and pollution of the examined river sections. Our results were in accordance with published data. For example, diatoms dominated in the mountain lakes of Spain [45] and the periphyton of the upper sections of alpine rivers, with severe conditions [46]. Cyanobacteria mainly developed under milder conditions of the lower sections of alpine rivers. Earlier, we revealed the influence of the altitude on the structure of plankton communities [12,47].

It is known that the species composition of algal communities varies depending on pH, salinity, temperature, oxygen conditions [42], and organic pollution of water bodies [38,41,48]. Toxic pollution of biotopes causes not only changes in the species composition of algal communities [49] but also the appearance of cells with morphological alterations [50].

The high sensitivity of algal communities to external factors determines their widespread use for assessing the quality of water in reservoirs [42,48,51,52]. One of the approaches that we used in this study is to assess the water quality of water bodies based on the environmental preferences of algal species [42]. The analysis showed that algae communities consisted mainly of benthic and plankton-benthic species (Figure 4), which are common for periphyton [53]. The ratio of indicator species in periphyton communities indicated favorable oxygen conditions, a relatively wide gradient of salinity, an alkaline water reaction, and a moderate level of organic pollution in the Arys River basin. The environmental preferences of the algae indicated weak or moderate levels of organic pollution in most parts of the Arys River basin.

The environmental preferences of the algal species common to each cluster generally coincided with the distribution of chemical parameters in different parts of the basin.

Transparency characterized a very low water quality (3–5 grades) of the greater part of the Arys River basin, which may be due mainly to natural causes. The specific color of water, including in the background sections of rivers (Table 2), is due to the erosion of loess, loess-like, and clayey rocks, which are characteristic of this region [54]. On the one hand, this worsens the conditions for the development of periphyton photosynthetic algae due to the water turbidity, but on the other hand, loess particles absorb pollutants from the water, slightly increasing its quality.

Despite intensive land use [7,8], the nutrient content in the water of the Arys River basin was relatively low (Table 4). Estimation of the level of organic pollution by the saprobity index SI calculated for periphyton communities (Figure 6f) generally corresponded to the data of chemical analysis. The permanganate index values (Figure 6b) also testified to the low amount of organic substances in the Arys River. Environmental mapping showed that the deterioration of water quality according to nitrite-nitrogen and nitrate-nitrogen content occurred mainly in the middle reaches of the Arys River (Figure 6c,d), the most used for agriculture and livestock (Figure 1). The improvement in water quality in the lower, less populated areas of the basin (Figure 6) indicated a good self-cleaning ability of the river ecosystem. It is obviously associated with the sorption properties of clay, which effectively binds heavy metals [55,56] and organic compounds [57]. It follows that a large number of suspended clay particles and, as a consequence, low transparency of water (Figure 6a) contributed to the natural purification of the Arys River water from incoming pollutants. Besides, the low level of organic pollution in the Arys basin is connected with the prevalence of clay soil, which is poor by organic matter [58].

The WESI index makes it possible to assess the general level of toxic effects on the autotrophic organisms of aquatic ecosystems [42]. Earlier, this index was successfully applied to assess the total level of toxic pollution of water bodies in various regions [59–64]. According to WESI values, the mountainous part of the Arys River basin was characterized by a low level of toxic pollution (Figure 7). The toxic pollution of the middle reaches of the Arys River, along with its tributaries, is due to the anthropogenic use of the territory, including the influence of industrial enterprises and agriculture. As with nitrogen compounds (Figure 6c,d), a decrease in toxic pollution downstream can be attributed to the sorbing properties of clay [57,58]. The lowest section of the Arys River, before it flows into the Syr Darya, was characterized by a deterioration in water quality in terms of the content of nitrogen compounds and the WESI index. One of the reasons for this may be the influx of heavy metals from the underlying rocks and from the catchment area since this region has a large deposit of polymetallic ores (Figure 1, POLY). Indirect evidence of this is the increase in the content of heavy metals in the Syr Darya River below the confluence of the Arys River [65].

The results obtained showed the complex nature of the interaction of natural and anthropogenic factors that determines the quality of water in the Arys River basin. Ecological mapping is one of the effective tools for assessing spatial heterogeneity of water quality and can be used in further monitoring of water bodies in this populated region.

## 7. Conclusions

A comprehensive assessment of the water quality of the Arys River basin is given based on biological and abiotic indicators. Cluster analysis revealed the uniqueness of the composition of phytoperiphyton communities of the examined river sections. With a low level of similarity (40%), the association of periphyton communities into clusters was associated with an altitudinal-climatic gradient, as well as nutrient content and pollution. The environmental preferences of the algae species common to each cluster generally coincided with the distribution of chemical variables in different parts of the basin. Mapping of abiotic and biological data revealed the worst water quality in the middle reaches of the Arys River as the most developed and populated part of the basin. We attribute the local improvement in water quality in the lower reaches of the Arys River to the nature of the underlying soils and the sorption of pollutants by clay suspended in water. A secondary deterioration in water quality according to the WESI index (an indicator of the effect of toxic pollution on autotrophs) in the section of the Arys River before it flows into the Syr Darya may be due to the influx of heavy metals from the underlying rocks and the catchment area. Thus, the studies showed a complex interaction of natural and anthropogenic factors causing changes in water quality in the Arys River basin.

**Author Contributions:** Conceptualization, E.G.K. and S.S.B.; Data curation, E.G.K.; Formal analysis, E.G.K., S.S.B. and S.M.R.; Investigation, E.G.K.; Methodology, E.G.K. and S.S.B.; Project administration, E.G.K. and S.S.B.; Resources, E.G.K.; Software, S.S.B.; Supervision, S.S.B.; Validation, S.S.B.; Visualization, S.S.B.; Writing—original draft, E.G.K., S.S.B. and S.M.R.; Writing—review and editing, E.G.K., S.S.B. and S.M.R.

**Funding:** This research received no external funding.

**Acknowledgments:** This work was partly supported by the Israeli Ministry of Aliya and Integration.

**Conflicts of Interest:** The authors declare no conflict of interest.

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
