# Peer review of "Ecological Mapping in Assessing the Impact of Environmental Factors on the Aquatic Ecosystem of the Arys River Basin, South Kazakhstan"

_diversity, doi:10.3390/d11120239_

Round 1

Reviewer 1 Report

The paper contains the results of assessing the water quality of 13 stations along the Arys River basin. Analyses were made based on the environmental mapping of two components - abiotic variables and biological factors, which were represented by phytoperiphytic communities. The Authors proved, using cluster analysis, that there was a distinctive composition of epiphytic communities, which were affected by the altitudinal-climatic gradient and the pollution of the main river and its tributaries as well as nutrient concentrations. The permanganate index, which  is the method applied to assess the water quality, showed generally low organic pollution of water, particularly in the middle part of the river basin.

The results of this study were briefly and efficiently presented, providing an assessment of the ecological state of the basin of main river Arys with the use of biotic and abiotic indicators. The whole manuscript is properly constructed and based on original data. It presents methods that can be used in ecological studies where we expect to identify the impact of the surrounding area on the inhabiting organisms. However, there are some shortcomings and deficiencies which should be remedied so that the whole paper will reach the satisfactory level.

Specific comments:

Abstract

In this section the results of the periphytic communities are lacking. Abiotic variables are widely presented while the biological description is quite poor. This should be added as the purpose of work also refers to biological variables.

Introduction

This chapter of the manuscript is properly constructed without any long or unnecessary descriptions. It clearly specifies the research object and the need to study the lake ecosystem. Only some minor remarks of an excessive vague character that appear in some parts of the text and minor errors require the authors' attention.

Line 30: insert human population density;

Line 36: slightly change the sentence: The plains of the river basin are prone to contamination caused by the use …;

Lines 40-41: slightly change the sentences: … Arys River has not yet been carried out, however, data on hydrochemical variables …;

Lines 46-47: slightly change the sentences: are unfavorable for the development of phytoplankton [16] and particularly zooplankton [17], which …..

In the final paragraph of this chapter – in the aim of the study it should be necessary to add some information whose parameters will be included in the assessment of ecological state. You should refer to the periphytic community as a biological variable and to certain abiotic properties.

Description of the study area

The Authors did not provide any morphological features of the main river such as its length, width and depth, although in the chapter Results (Table 2) there is some information on the depth at each sampling station. In this chapter at least some ranges of morphometric parameters should be found.

Material and Methods

I do not know why there were 28 samples collected from 13 stations. Were samples collected only on one day? Were some sample stations examined better than others? What was the reason for collecting more samples on certain stations and less samples at others? Please explain.

Line 96: insert the full stop: … June 2016. Black squares are cities …

Were periphytic samples sedimented at the laboratory?

Results

In the descriptions of the tables I would recommend adding the information referring to certain sampling stations, at least in brackets.

Why is the chapter referring to epiphyton named: Brief description of phytoperiphyton? It should rather be e.g. Characteristics of phytoperiphyton. This suggests that a longer description will be given too.

The Authors use taxa or species interchangeably. It is biologically precisely defined. Therefore, the nomenclature appropriate to the obtained taxonomic results should be used throughout the whole text. Genera names should be in italics.

Line 139: Green and charophyte algae that are mostly plankton inhabitants. Charophytes are not planktonic organisms!

Discussion

In the discussion part, the Authors should definitely put more stress on presenting and discussing the data on the species composition of periphyton communities. This aspect was neglected in this part of the manuscript (the same refers to the Abstract as mentioned earlier).

I would also advise the authors’ to re-write parts of the Discussion because there are some speculations, e.g.  about the influx of heavy metals from the underlying rocks and the catchment area. There may be some literaturę available on this subject from the Syr Darya basin. This would give some suport to such a statement.

English

The English of the manuscript is good, however, the text should be read for language checking before final submission as several mistakes occurred.

Author Response

Dear Editor,

Thank you and the Reviewers for comments. Please consider corrected ms in respect of the comments and our answers to the comments point by point, below.

With best regards,

Prof Sophia Barinova

Reviewer 1

Notes

Answers

Abstract In this section the results of the periphytic communities are lacking. Abiotic variables are widely presented while the biological description is quite poor. This should be added as the purpose of work also refers to biological variables.

corrected

Introduction Line 30: insert human population density;

Line 36: slightly change the sentence: The plains of the river basin are prone to contamination caused by the use …;

Lines 40-41: slightly change the sentences: … Arys River has not yet been carried out, however, data on hydrochemical variables …;

Lines 46-47: slightly change the sentences: are unfavorable for the development of phytoplankton [16] and particularly zooplankton [17], which …..

In the final paragraph of this chapter – in the aim of the study it should be necessary to add some information whose parameters will be included in the assessment of ecological state. You should refer to the periphytic community as a biological variable and to certain abiotic properties.

Corrected

Description of the study area

The Authors did not provide any morphological features of the main river such as its length, width and depth, although in the chapter Results (Table 2) there is some information on the depth at each sampling station. In this chapter at least some ranges of morphometric parameters should be found.

corrected

Material and Methods

I do not know why there were 28 samples collected from 13 stations. Were samples collected only on one day? Were some sample stations examined better than others? What was the reason for collecting more samples on certain stations and less samples at others? Please explain.

Line 96: insert the full stop: … June 2016. Black squares are cities …

Were periphytic samples sedimented at the laboratory?

Corrected.

A total of 28 samples from different substrates (stones, water plants) were taken. Two samples of phytoperiphyton most often were collected from each station. Phytoperiphyton samples were collected taking into account visible differences in submerged substrates.

The algal samples were obtained by scratching of periphyton, putted to the 15 ml plastic tubes, and fixed in 4% of neutral formaldehyde solution

Results

In the descriptions of the tables I would recommend adding the information referring to certain sampling stations, at least in brackets.

Why is the chapter referring to epiphyton named: Brief description of phytoperiphyton? It should rather be e.g. Characteristics of phytoperiphyton. This suggests that a longer description will be given too.

The Authors use taxa or species interchangeably. It is biologically precisely defined. Therefore, the nomenclature appropriate to the obtained taxonomic results should be used throughout the whole text. Genera names should be in italics.

Line 139: Green and charophyte algae that are mostly plankton inhabitants. Charophytes are not planktonic organisms!

corrected

Discussion

In the discussion part, the Authors should definitely put more stress on presenting and discussing the data on the species composition of periphyton communities. This aspect was neglected in this part of the manuscript (the same refers to the Abstract as mentioned earlier).

I would also advise the authors’ to re-write parts of the Discussion because there are some speculations, e.g.  about the influx of heavy metals from the underlying rocks and the catchment area. There may be some literaturę available on this subject from the Syr Darya basin. This would give some suport to such a statement.

corrected

Reviewer 2 Report

The purpose of the work is to assess the impact of environmental factors on the periphyton in a river. Using environmental mapping and cluster analysis, authors for the first time conducted a comprehensive assessment of the Arys river in South Kazakhstan. The authors revealed specific periphyton communities with the altitudinal gradient, the nutrient content, and the pollution of the examined rivers.

The work is interesting and will be interesting to a wide circle of hydrobiologists, but publication is possible only after the revision of the manuscript.

The following are section notes.

Section “Introduction”

Lines 30-31: Maybe it would be valuable to place your interesting study in a broader context. There is a number of works concerning abiotic factors and biota in relation to land use.

e.g. 

Soranno, P. A., Hubler, S. L., Carpenter, S. R., & Lathrop, R. C. (1996). Phosphorus loads to surface waters: a simple model to account for spatial pattern of land use. Ecological Applications, 6(3), 865-878.

Lines 45-47 – Maybe it would be worth to mention some papers that treat phytoplankton and zooplankton as an indicator for land use characterization in mountain rivers.

e.g.

Rimet, F., Gomà, J., Cambra, J., Bertuzzi, E., Cantonati, M., Cappelletti, C., ... & Tison, J. (2007). Benthic diatoms in western European streams with altitudes above 800 M: characterisation of the main assemblages and correspondence with ecoregions. Diatom research, 22(1), 147-188.

Sługocki, Ł., Czerniawski, R., Kowalska-Góralska, M., Senze, M., Reis, A., Carrola, J., & Teixeira, C. (2019). The Impact of Land Use Transformations on Zooplankton Communities in a Small Mountain River (The Corgo River, Northern Portugal). International journal of environmental research and public health, 16(1), 20.

Section „Material and Methods”

Lines 83-84: Could you estimate the size of samples (volume, scratching time, area size?). There was the same sampling effort in each station? Those questions, in my opinion, are especially important because you have not an equal number of samples in all stations (13 stations and 28 samples).

Lines 84-86: Please indicate devices that you used for measuring each parameter.

Line 113: Could you indicate what calculation have you used for measuring similarities?

Have you measured the water current velocity at each station? The SD depth value could be correlated with high water current. In low current velocity sedimentation would be high therefore transparency in that station could be higher.

Section „Results”

The symbol of Celsius degree value should be like this “°C” instead of “◦C”

Section „References”

Line 361 – there is a lack of numbering for that reference. Please check the references again.

Author Response

Dear Editor,

Thank you and the Reviewers for comments. Please consider corrected ms in respect of the comments and our answers to the comments point by point, below.

With best regards,

Prof Sophia Barinova

Reviewer 2

Lines 30-31: Maybe it would be valuable to place your interesting study in a broader context. There is a number of works concerning abiotic factors and biota in relation to land use.

corrected

Lines 45-47 – Maybe it would be worth to mention some papers that treat phytoplankton and zooplankton as an indicator for land use characterization in mountain rivers.

corrected

Rimet, F., Gomà, J., Cambra, J., Bertuzzi, E., Cantonati, M., Cappelletti, C., ... & Tison, J. (2007). Benthic diatoms in western European streams with altitudes above 800 M: characterisation of the main assemblages and correspondence with ecoregions. Diatom research, 22(1), 147-188. перифитон

It is the article about bentic algae but not about phytoplankton.

Soranno, P. A., Hubler, S. L., Carpenter, S. R., & Lathrop, R. C. (1996). Phosphorus loads to surface waters: a simple model to account for spatial pattern of land use. Ecological Applications, 6(3), 865-878

added

Sługocki, Ł., Czerniawski, R., Kowalska-Góralska, M., Senze, M., Reis, A., Carrola, J., & Teixeira, C. (2019). The Impact of Land Use Transformations on Zooplankton Communities in a Small Mountain River (The Corgo River, Northern Portugal). International journal of environmental research and public health, 16(1), 20

added

Lines 83-84: Could you estimate the size of samples (volume, scratching time, area size?). There was the same sampling effort in each station? Those questions, in my opinion, are especially important because you have not an equal number of samples in all stations (13 stations and 28 samples).

corrected

Lines 84-86: Please indicate devices that you used for measuring each parameter.

Corrected. Most water samples were analyzed in the laboratory.  The temperature and pH values of the surface layers of water were measured in parallel with sampling by HANNA HI 9813-0. The transparency of the water was measured using a Secchi disk. Coordinate referencing of the stations was done by Garmin eTrex GPS-navigator.

Line 113: Could you indicate what calculation have you used for measuring similarities?

corrected

Have you measured the water current velocity at each station? The SD depth value could be correlated with high water current. In low current velocity sedimentation would be high therefore transparency in that station could be higher. 

You are right, but unfortunately, we didn’t measure the water current velocity.

Line 361 – there is a lack of numbering for that reference

corrected
